# A Combined Model of SARIMA and Prophet Models in Forecasting AIDS Incidence in Henan Province, China

**DOI:** 10.3390/ijerph19105910

**Published:** 2022-05-12

**Authors:** Zixiao Luo, Xiaocan Jia, Junzhe Bao, Zhijuan Song, Huili Zhu, Mengying Liu, Yongli Yang, Xuezhong Shi

**Affiliations:** Department of Epidemiology and Biostatistics, College of Public Health, Zhengzhou University, Zhengzhou 450001, China; lzxiao0206@163.com (Z.L.); jxc@zzu.edu.cn (X.J.); baojz@zzu.edu.cn (J.B.); zjsong178@163.com (Z.S.); zhuhuili39@163.com (H.Z.); 18063131171@163.com (M.L.)

**Keywords:** AIDS, SARIMA model, Prophet model, L1-norm

## Abstract

Acquired immune deficiency syndrome (AIDS) is a serious public health problem. This study aims to establish a combined model of seasonal autoregressive integrated moving average (SARIMA) and Prophet models based on an L1-norm to predict the incidence of AIDS in Henan province, China. The monthly incidences of AIDS in Henan province from 2012 to 2020 were obtained from the Health Commission of Henan Province. A SARIMA model, a Prophet model, and two combined models were adopted to fit the monthly incidence of AIDS using the data from January 2012 to December 2019. The data from January 2020 to December 2020 was used to verify. The mean square error (MSE), mean absolute error (MAE), and mean absolute percentage error (MAPE) were used to compare the prediction effect among the models. The results showed that the monthly incidence fluctuated from 0.05 to 0.50 per 100,000 individuals, and the monthly incidence of AIDS had a certain periodicity in Henan province. In addition, the prediction effect of the Prophet model was better than SARIMA model, the combined model was better than the single models, and the combined model based on the L1-norm had the best effect values (MSE = 0.0056, MAE = 0.0553, MAPE = 43.5337). This indicated that, compared with the L2-norm, the L1-norm improved the prediction accuracy of the combined model. The combined model of SARIMA and Prophet based on the L1-norm is a suitable method to predict the incidence of AIDS in Henan. Our findings can provide theoretical evidence for the government to formulate policies regarding AIDS prevention.

## 1. Introduction

Acquired immune deficiency syndrome (AIDS) is caused by the human immune deficiency virus (HIV). According to the latest data of the Joint United Nations Programme on HIV/AIDS, approximately 37.6 million people were living with HIV and AIDS worldwide at the end of 2020 [1]. By the end of 2018, an estimated 1,250,000 people were living with HIV in China, and 80,000 new infections were reported in 2018 [2]. Henan is one of the provinces hit hardest by HIV and AIDS in China [3,4]. In the past, AIDS patients mostly appeared in paid blood donors in central China, especially in Henan province. Plasma marketing in Henan province in 1994–1995 lead to the rapid emergence of a large number of AIDS cases in the early 2000s [5]. Henan province ranks first in terms of population. Although China has responded with the prevention and treatment of AIDS, there is still a certain number of undetected HIV infections and continuous spread in society every year, which has led to a serious AIDS epidemic in Henan province [6]. The number of AIDS patients in Henan province ranked first in China in 2015, and 65,896 people were living with HIV and AIDS in Henan province in October 2020 [7]. At present, the medical field has not developed specific drugs to cure AIDS or an effective vaccine to prevent AIDS. Due to the latter, developing scientific and effective AIDS prevention and control strategies has become a top priority to curb the AIDS epidemic. Therefore, it is important to establish a mathematical model of AIDS epidemic prediction so as to understand the process of the AIDS epidemic, explore its epidemic characteristics and develop law, and seek the optimal strategy for its prevention and control [8].

Over the past few years, mathematical models have been used to successfully predict the incidence of HIV and AIDS [9,10]. At present, the most common method used to predict disease incidences is the autoregressive integrated moving average (ARIMA) model, which is been widely used for the prediction of timeseries data, such as the incidence of human brucellosis [11] and COVID-19 [12]. However, ARIMA models have one major limitation of pre-assumed linearity [13]. In most cases, nonlinear structures are adopted during timeseries analyses, as adequate results cannot be obtained from linear models. On the contrary, a Prophet model does not need to specify a detailed model, and it simultaneously simulates multiple seasons through a generalized additive model. It adopts a Bayesian-based curve-fitting method to smooth and forecast timeseries data, and it has a fast and robust fitting process for large outliers, missing values, and dramatic changes [14]. To date, Prophet model has been used in many disciplines, including infectious diseases such as COVID-19 [15,16], but few studies have applied it to AIDS. In the present study, we have built both Seasonal Autoregressive Integrated Moving Average (SARIMA) and Prophet models because time series data for AIDS had both linear and non-linear characteristics.

It is universally agreed that combining different models can increase the chance of capturing various data features and improve prediction accuracy [17,18]. More recently, combined forecasting models have been extensively applied in various fields with high predictive performance, including air quality [19] and influenza [20]. However, the parameter estimation methods of proposed models have mainly been based on the minimum L2-norm of the prediction error vector, that is, the combination prediction model is based on the minimum sum of the squares of the prediction error. However, the prediction error becomes enlarged or reduced after the prediction error is squared [21]. Considering the defect of the L2-norm, it is necessary to introduce the prediction error vector L1-norm index, which uses the sum of the absolute values of the prediction errors. Its robustness is better than prediction residual error sum of squares, especially when there are abnormal, extreme values in the data [22,23]. Wang et al. proposed a combined model to predict air pollutant concentration. Based on the L1-norm, the model performed a weighted combination of the prediction results of three single models (extreme learning machine, Elman neural network, and support vector machine), and the results showed that the proposed combined model had a stable prediction performance [24]. Therefore, we propose a combined model of the SARIMA and Prophet models to predict AIDS incidence based on the L1-norm. We compare the prediction effects to explore which is the most precise model for AIDS incidence prediction in Henan province. The results provide reference information for AIDS prevention and intervention in Henan province.

## 2. Methods

### 2.1. Data Sources

The incidence data of AIDS in Henan from January 2012 to December 2020 were obtained from the Health Commission of Henan Province (http://wsjkw.henan.gov.cn/, accessed on 1 April 2021). The monthly incidences of AIDS in Henan province from January 2012 to December 2019 were set as the training set, and the incidences of AIDS from January to December 2020 were set as the test set. A SARIMA model, a Prophet model, and two combined models were adopted to fit the monthly incidence of AIDS by using the data from the training set. The forecasting performances of the four fitted models were verified by using the data from the test set. The technical roadmap is shown in Figure 1.

### 2.2. SARIMA Model

ARIMA model consists of three parts: autoregression (p), the degree of difference (d), and the order of moving average (q) [25]. For seasonal trends, a SARIMA model combines nonseasonal and seasonal components and can be specified as SARIMA (p, d, q) × (P, D, Q)s, where p, d, and q refer to the orders of the nonseasonal autoregressive (AR), nonseasonal differencing, and nonseasonal moving average (MA) parts of the model. P, D, and Q refer to the orders of the seasonal AR, seasonal differencing, and seasonal MA parts of the model, and the subscripted letter “s” is the length of the seasonal period [26]. In this study, the incidence of AIDS varied in the annual cycle, so s = 12.

A timeseries modeling approach involves the following three steps: model identification, parameter estimation, and diagnostic checking. Firstly, if it is necessary, appropriate differencing of the series is performed to achieve stationarity and normality. We used an augmented Dickey–Fuller (ADF) unit root test to estimate whether the timeseries was stationary or not. Secondly, the temporal correlation structure of the transformed data is identified by examining its autocorrelation (ACF) and partial autocorrelation (PACF) functions. In addition, the values of p, P and q, Q are finally determined by considering that smaller Akaike information criterion (AIC) and Bayesian information criterion (BIC) values correspond to a higher prediction accuracy. Finally, in order to test the normality of the SARIMA residuals, a white noise test is conducted in the residual series [27].

### 2.3. Prophet Model

A Prophet model is a data prediction tool for the timeseries of Facebook. It was introduced by Taylor and Letham and it is available in packages such as Python and R [14]. It can almost automatically predict the future trend of a timeseries. In addition, the model deals with the case of outliers in a time series and it also deals with some missing values. A Prophet model adopts a curve-fitting method based on Bayes to smooth and forecast timeseries data, so the results that need to be predicted can be obtained faster. In general, timeseries prediction or data analysis can use this algorithm to predict the trend of future timeseries. The formulation of a Prophet model is similar to a generalized additive model, including trend, seasonality, and holidays: *y*(*t*) = *g*(*t*) + *s*(*t*) + *h*(*t*) + *ε_t_*
where *g*(*t*) is the trend function representing nonperiodic changes in timeseries values, and *s*(*t*) represents periodic changes (for example, weekly and annual seasonality). *h*(*t*) represents the effects of holidays that occur on potentially irregular schedules over one or more days. *ε_t_* is the error term and was assumed to be normally distributed in this study.

For a trend model, it involves fitting a piecewise linear curve or a nonlinear saturating growth model. This sort of growth is typically modeled using a logistic growth model, which, in its most basic form, is:$$g(t)=\frac{C}{1+\exp\left(-k\left(t-m\right)\right)}$$
where *C* indicates the carrying capacity, *k* is the growth rate, and *m* represents an offset parameter. Both the carrying capacity and rate of growth are not constant. By altering the parameter rate, the flexibility of the model can be controlled [28].

In this study, the change points were automatically selected, the number of change points was set as 25, and the carrying capacity of the logistic growth model was set as 8.5. We set the interval width as 0.85 and the number of uncertainty samples as 1000.

### 2.4. The Combined Model Based on L1-Norm

Supposing the observed values of an index are {$x_{t}$, *t* = 1, 2, …, N}, there are *m* feasible single prediction methods to forecast it, where $x_{it}$ is the predicted value of *i*th method at time *t*, *i* = 1, 2, …, *m*, *t* = 1, 2, …, *N*. When {$l_{i}$, *i* = 1, 2, …, *m*} is the weighting coefficient of *m* single prediction in the combined prediction model, it satisfies normality and non-negativity:(1)$$\sum_{i=1}^{m}l_{i}=1,\;l_{i}\;\geq \;0,\;i=1,\;2,\;\ldots ,\;m$$

We first considered $\hat{x_{t}}$, applying weighted geometric mean:(2)$$\hat{x_{t}}=\prod_{i=1}^{m}x_{it}^{l_{i}},t=1,\;2,\;\ldots ,\;N$$
(3)$$\ln\hat{x_{t}}=\sum_{i=1}^{m}l_{i}lnx_{it},t=1,\;2,\;\ldots ,\;N$$
where $\hat{x}$ is the weighted geometric average of $x_{t}$.

Supposing $e_{t}$ is the logarithmic error between the combined predicted value and the corresponding actual value at time *t* yields the following:(4)$$e_{it}=lnx_{t}-lnx_{it}$$
(5)$$e_{t\;}=lnx_{t}-\;ln\hat{x_{t}}=\sum_{i=1}^{m}l_{i}\left(lnx_{t}-lnx_{it}\right)=\sum_{i=1}^{m}l_{i}e_{it}$$
where $e_{it}$ is the logarithmic error between the actual value at time *t* and the corresponding predicted value of the *i*th single model, *i* = 1, 2, …, *m*, *t* = 1, 2, …, *N*.

*F* is the logarithmic error between the combined prediction model and the actual value of the index based on the L1-norm, and $F_{i}$ is defined as the logarithmic error between the predicted value and the corresponding actual value of the *i*th single model, yielding:(6)$$F=\sum_{t=1}^{N}\left|e_{t}\right|=\sum_{t=1}^{N}|\sum_{i=1}^{m}l_{i}e_{it}|$$
(7)$$F_{i}=\sum_{t=1}^{N}\left|e_{it}\right|,i=1,\;2,\;\ldots ,\;m$$
where *F* is the function of the weighting coefficient vector *L* = ${\left(l_{1},l_{2},\ldots ,l_{m}\right)}^{T}$ of various prediction methods, which can be denoted as *F*(*L*).

In the ideal case, if there is no prediction error, *F*(*L*) = 0. However, predicted error is inevitable. The smaller *F*(*L*) is, the closer the combined prediction value is to the actual value of the index, and the more accurate and effective the combined model. Therefore, model (1) was expressed as:$$\min\;F\left(L\right)=\overset{N}{\underset{t=1}{\sum }}|\overset{m}{\underset{i=1}{\sum }}l_{i}e_{it}|\;sꞏt\{\begin{array}{c} \sum_{i=1}^{m}l_{i}=1,\\ l_{i}\geq 0,\;i=1,\;2,\;\ldots ,\;m \end{array}$$

In this study, *m* = 2, *N* = 12, $l_{1}$ was the weighting coefficient of the SARIMA model and $l_{2}$ was the weighting coefficient of the Prophet model.

### 2.5. Model Evaluation

The Akaike information criterion (AIC) and Bayesian information criterion (BIC) were used to screen parameters of the SARIMA model. The SARIMA model with the minimum AIC and BIC was the most suitable one. The models were estimated using the mean square error (MSE), mean absolute percentage error (MAPE), and mean absolute error (MAE), and the model with the smallest values of these indices was identified as optimal [10].

### 2.6. Data Processing and Analysis

R software (version 3.6.2, R Foundation for Statistical Computing, Vienna, Austria) was adopted to develop the SARIMA model and the Prophet model. In addition, LINGO software (version 15.0, Lindo System Inc., Chicago, IL, USA) was adopted to create the combined model. The significant level was 0.05. 

## 3. Results

### 3.1. Trends of AIDS in Henan Province

The timeseries data covered 108 months, from January 2012 to December 2020. As shown in Figure 2, the monthly incidence fluctuated from 0.05 to 0.50 per 100,000 individuals, and the monthly incidence of AIDS in Henan province had a certain periodicity. The peaks of the disease occurred mainly in December, while the low peaks occurred mainly in January or February. There was a sudden decline in the monthly incidence of AIDS in January 2020 and a record low in February 2020.

### 3.2. SARIMA Models

For SARIMA, the ADF test indicated that the original series was stationary (Dickey–Fuller  = − 3.796, *p*  < 0.05), which did not need trend differencing ACF and PACF graphs were used to estimate the parameter ranges of p, P and q, Q. We found that the plots of the original series displayed slow decay at the seasonal lags (Figure 3a). Therefore, at lag-12 (subtracting the observations after a lag of 12 periods) differencing was used to remove seasonality features. The sequence of one-order seasonal difference was stable (Figure 3b). Then, some candidate SARIMA models were assessed to forecast future values based on the previously observed values (Table 1). Further, of all the tested models shown in Table 1, the SARIMA(1,0,1)(0,1,1)[12] model was found to best fit the data, which had the lowest Akaike information criterion (AIC = −253.67) and Bayesian information criterion (BIC = −244.00). This model also passed the Ljung–Box Q Test (*p* = 0.420); the testing results of the estimated parameters were all statistically significant (*p* < 0.05).

Finally, we predicted the monthly incidence of AIDS from January to December in 2020 with the SARIMA(1,0,1)(0,1,1)[12] model. The results are shown in Table 2.

### 3.3. Prophet Model

The prophet model was automatically fitted with the incidence rates in Henan province from January 2012 to December 2019, and then the AIDS incidence rates from January to December 2020 were predicted. The predicted results are shown in Table 2, and the fitting prediction curve is shown in Figure 4. The results showed that the monthly incidence of AIDS was seasonal in Henan province.

The decomposed components of the monthly incidence of AIDS included the effect of trend and the yearly seasonality (Figure 5). An increasing trend in the reported incidence of AIDS was observed from 2012 to 2020. For yearly seasonality, an apparent local maximum appeared in September, and an apparent local minimum appeared in November.

### 3.4. The Combined Model

For the combined model based on the L2-norm, we obtained $l_{1}$ = 0.548 and $\;l_{2}$ = 0.452. For the combined model based on the L1-norm, we obtained $l_{1}$ = 0.4587 and $\;l_{2}$ = 0.5417. The prediction values of the combined models are shown in Table 3. 

According to model (1), the minimum log error of the combined model based on the L1-norm was *F*($l_{1}$,$\;l_{2}$) = 3.5479, while the logarithmic error of the SARIMA model was $F_{1}$ = 4.0063 and that of the Prophet model was $F_{2}$ = 3.8331. The combinatorial model based on the L1-norm was the optimal combinatorial model, as it resulted in *F*($l_{1}$,$\;l_{2}$) < min{$F_{1}$,$\;F_{2}$}.

### 3.5. Model Evaluation

Compared with the other models, the effect values of the combined model based on L1 norm were all lower (Table 3). It showed that the prediction effect of the combined model based on L1 was the best.

## 4. Discussion

In this study, we focused on monthly incidence data for AIDS from 2012 to 2020 in Henan province, China. The results showed that the incidence of AIDS in Henan province was relatively stable. We built four models for analyzing the timeseries: a SARIMA model, a Prophet model, and two combined models of SARIMA and Prophet. The predictive abilities of the four models were compared and it was discovered that the combined model based on the L1-norm proposed in this study had the best model effect values and was superior to the other models in predicting effect. These findings indicated the potential value of the combined model based on the L1-norm in forecasting short-term AIDS incidence in Henan province. The model can provide reference for AIDS prevention and intervention in Henan province.

The Henan provincial government has done an excellent job in the prevention and treatment of AIDS during a transfer of AIDS carriers to a new AIDS population in Henan province. It warned us that the AIDS epidemic cannot be ignored. The Henan government has not slackened on AIDS in recent years. To implement China’s 13th Five-Year Action Plan for HIV/AIDS Containment and Prevention, the Henan University AIDS Prevention and Control Alliance was established in 2018 [7]. In the face of the impact of COVID-19, we still need to pay attention to the prevention and control of AIDS [29,30]. It requires us to go further in AIDS prevention, treatment, and research and to raise public awareness of the growing threat posed by infectious diseases. In addition, AIDS is not a seasonal infection. Still, the findings suggested some seasonality in the AIDS incidence data reported through the Center for Disease Control and Prevention (CDC): the rate of new infections is low in January and February and this can be attributed to the influence of the annual Spring Festival, which falls in late January or early February. During the Spring Festival, national and provincial CDCs and most hospitals and clinical laboratories operate with limited capacity, and people’s willingness to seek medical treatment falls, resulting in low AIDS infection records. Some other studies have also supported the latter phenomenon [31].

Different models have their own merits and faults. Based on historical AIDS incidence data, our study used timeseries analysis methods to establish SARIMA(1,0,1)(0,1,1)[12], a Prophet model, and two combined models for forecasting the monthly incidence of AIDS in Henan province. We found that the prediction effect of the Prophet model was better than the SARIMA model. We know that the SARIMA model combines the advantages of the two methods of regression analysis and moving average, which can analyze seasonal timeseries [32]. A previous study found that the most appropriate ARIMA models for HIV incidence in 2015 and 2016 in Guangxi, China, were ARIMA(1,1,2)(0,1,2)[12] and ARIMA(2,1,0)(1,1,2)[12], respectively [8]. A limitation of the SARIMA model is that it is easy to overfit with poor generalization ability when processing daily data, and it is more suitable for linear models [13]. However, the monthly incidences of AIDS had both linear and nonlinear characteristics, and there was a missing value, which meant some information may not have been captured by the SARIMA model. Adopting a generalized additive model formulation, the Prophet model is fast in its fitting procedure and robust for large outliers, missing values, and dramatic changes. The Prophet model automatically fitted the data, and the interpolation of missing values was not required [14], so it produced better results. This may explain why the performances of the SARIMA model were not as accurate as those of the Prophet model, which was consistent with a previous study [33]. 

Previously, Adesoye Idowu Abioye et al. [34] used an ARIMA and Prophet model to fit and predict the COVID-19 cases in Nigeria from September to October 2020. On this basis, we built combined models to predict the incidence of AIDS in Henan province. The results showed that the prediction effects of the combined models were better than those of the two single models. Firstly, the combined models could combine the characteristics of a single model to capture more data information, and they were more suitable for fitting data with both linear and nonlinear characteristics, such as AIDS incidence. Secondly, the reasonable distribution of the weight of a single model may result in an ensemble with more accurate and lower variance [17]. To date, many related studies have shown that a combined model is better than a single model. For example, Grzegorz Dudek [35] believed that a hybrid residual dilated long short-term memory and exponential smoothing model was more competitive. Li [36] found that a combined model of modified linear ARIMA and modified nonlinear ARIMA improved the single models. Furthermore, we found that, compared with the L2-norm, the L1-norm improved the prediction accuracy of the combined model. The combined model based on the L2-norm was based on the minimum sum of the squares of the prediction error. However, the prediction error was enlarged or reduced after the prediction error was squared (that is, if the absolute value of the prediction absolute error was greater than 1, it was larger after it was squared; if the absolute value of the predicted relative error was less than 1, it was smaller after being squared). The combined model based on the L1-norm used the sum of the absolute values of the prediction errors. Based on the L1-norm, the model was sparse and regularized. The L1-norm decreased very quickly for small weights and slowly for large weights [24]. Overall, the combined forecasting model of SARIMA and Prophet models based on the L1-norm was an appropriate way for predicting the incidence of AIDS in Henan province. With the help of the combined model, it is reasonable for the government to allocate health resources to control the epidemic efficiently.

This study had several limitations. First, since the analyzed data was the monthly incidence data of AIDS, the decomposition components of the Prophet model only included the trend effect and yearly effect, which could not reflect the weekly effect and short-term holiday effect. Second, the timeseries prediction model was adopted without considering factors affecting the incidence of AIDS, such as age, gender, social status, epidemic variation, and humanity. Stratified analysis is essential for a better understanding of AIDS epidemiology. In the future, we can explore other suitable models (such as support vector regression, exponential smoothing models, and machine learning) for predicting AIDS in combination with epidemiological data and socio-economic determinants [37,38].

## 5. Conclusions

The combined forecasting model of the SARIMA and Prophet models based on the L1-norm was an appropriate way for predicting the incidence of AIDS in Henan province. The results showed that the combined model was suggested for use in AIDS surveillance, which provide evidence for the government to formulate policies by providing estimates on AIDS incidence trends in Henan, China.

## Figures and Tables

**Figure 1 ijerph-19-05910-f001:**
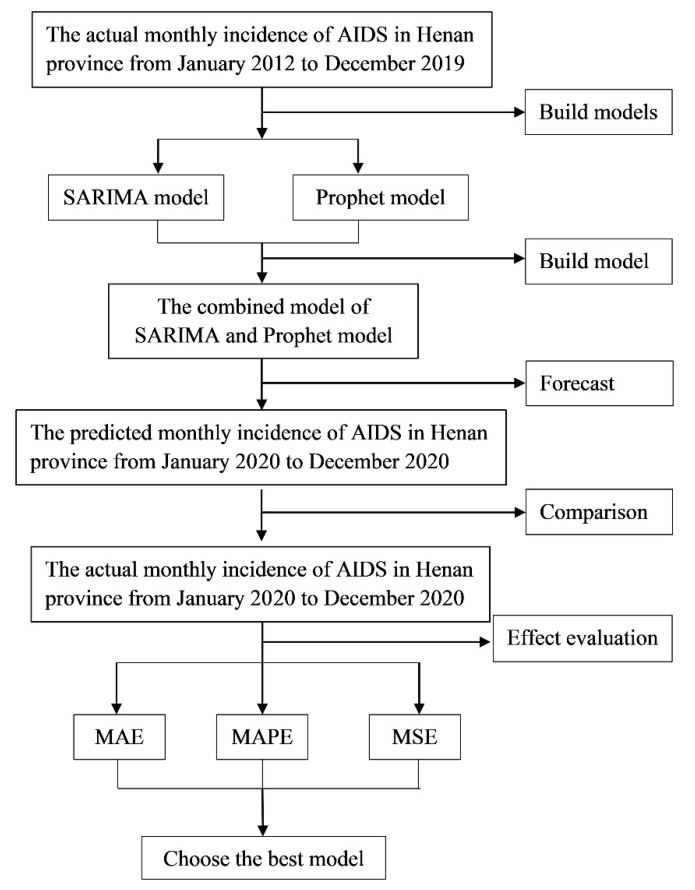
Technical roadmap.

**Figure 2 ijerph-19-05910-f002:**
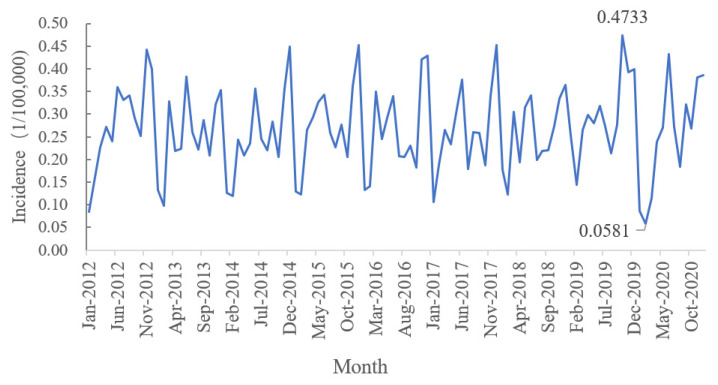
The timeseries diagram of monthly incidence of AIDS in Henan province from 2012−2020.

**Figure 3 ijerph-19-05910-f003:**
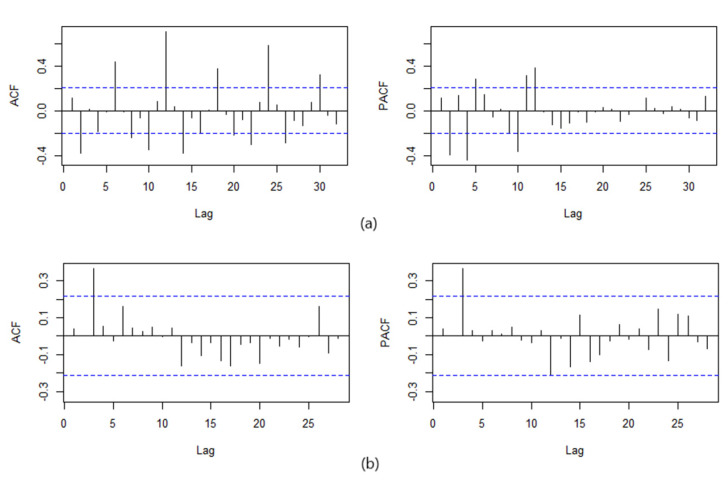
The ACF and PACF plots of monthly incidence of AIDS in Henan province from 2012−2019: (**a**) plots before seasonal difference and (**b**) plots of one-order seasonal difference.

**Figure 4 ijerph-19-05910-f004:**
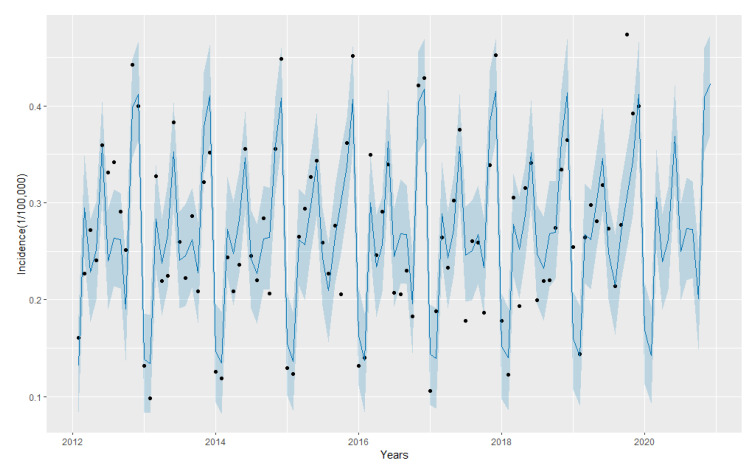
Predicted new infection rate of AIDS from January 2012 to December 2020 by Prophet model. The black dots represent the observed values, the blue line represents the fitted or predicted data of Prophet model, and the shadow area represents 95% confidence intervals.

**Figure 5 ijerph-19-05910-f005:**
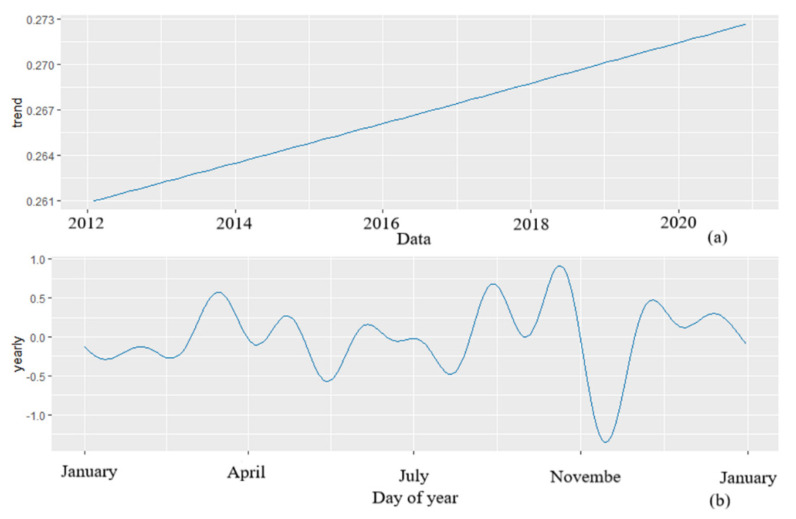
The decomposed components of the monthly incidence of AIDS in Henan province, China: (**a**) the effect of trend and (**b**) the effect of the yearly seasonality.

**Table 1 ijerph-19-05910-t001:** Comparison of candidate SARIMA models.

Model	Estimate	*p*-Value	Ljung–Box Q Test			AIC	BIC	RMSE	MAPE
			Statistics	DF	*p*-Value				
SARIMA(0,1,1)(0,1,0)[12]			19.42	17	0.305	−240.68	−235.87	0.050	14.008
MA1	−0.868	<0.001							
SARIMA(0,1,1)(1,1,2)[12]			14.96	15	0.454	−240.51	−228.48	0.046	12.249
MA1	−0.882	<0.001							
SAR1	−0.780	<0.001							
SMA1	0.567	0.046							
SMA2	−0.433	0.044							
SARIMA(1,0,1)(0,1,0)[12]			19.40	16	0.249	−249.89	−242.63	0.048	12.411
AR1	−0.777	<0.001							
MA1	1.000	<0.001							
SARIMA(1,0,1)(0,1,1)[12]			15.45	15	0.420	−253.67	−244.00	0.045	11.888
AR1	−0.751	<0.001							
MA1	1.000	<0.001							
SMA1	−0.398	0.019							
SARIMA(1,0,1)(1,1,0)[12]			16.85	15	0.328	−252.53	−242.85	0.046	12.178
AR1	−0.746	<0.001							
MA1	1.000	<0.001							
SAR1	−0.297	0.025							
SARIMA(1,0,1)(1,1,1)[12]			13.26	14	0.506	−252.07	−239.98	0.045	11.754
AR1	−0.759	<0.001							
MA1	1.000	<0.001							
SAR1	0.289	0.477							
SMA1	−0.688	0.084							
SARIMA(2,0,2)(0,1,0)[12]			18.18	16	0.313	−247.64	−235.55	0.047	12.780
AR1	−1.435	<0.001							
AR2	−0.925	<0.001							
MA1	1.581	<0.001							
MA2	1.000	<0.001							
SARIMA(2,0,2)(0,1,1)[12]			16.67	15	0.339	−251.68	−237.17	0.045	12.266
AR1	−1.067	<0.001							
AR2	0.950	<0.001							
MA1	1.907	<0.001							
MA2	0.830	<0.001							
SMA1	−0.500	<0.001							
SARIMA(3,0,0)(0,1,0)[12]			15.10	17	0.588	−256.66	−246.98	0.045	12.564
AR1	0.062	0.536							
AR2	−0.018	0.859							
AR3	0.469	<0.001							

AIC: Akaike information criterion; BIC: Bayesian information criterion; RMSE: root mean squared error; MAPE: mean absolute percent error; DF: degree of freedom.

**Table 2 ijerph-19-05910-t002:** Comparison of predicted values and actual values from January to December 2020 (per 100,000 population).

Time	Actual Value	Predicted Value			
		SARIMA(1,0,1)(0,1,1)[12]	Prophet Model	Combined Model Based on L2-Norm	Combined Model Based on L1-Norm
January-2020	0.087	0.235	0.168	0.205	0.199
February-2020	0.058	0.126	0.142	0.133	0.135
March-2020	0.114	0.289	0.305	0.296	0.298
April-2020	0.238	0.255	0.239	0.248	0.246
May-2020	0.270	0.298	0.262	0.282	0.279
June-2020	0.432	0.326	0.369	0.345	0.349
July-2020	0.273	0.247	0.250	0.249	0.249
August-2020	0.183	0.217	0.274	0.243	0.248
September-2020	0.322	0.262	0.273	0.267	0.268
October-2020	0.269	0.379	0.201	0.299	0.283
November-2020	0.382	0.375	0.409	0.390	0.393
December-2020	0.387	0.397	0.423	0.409	0.411

**Table 3 ijerph-19-05910-t003:** Effect evaluation of models.

Model	MSE	MAE	MAPE
SARIMA(1,0,1)(0,1,1)[12]	0.0073	0.0657	47.8470
Prophet Model	0.0060	0.0602	44.8336
Combined Model based on L2-norm	0.0060	0.057	44.1950
Combined Model based on L1-norm	0.0056	0.0553	43.5337

MSE: the mean square error; MAE: mean absolute error; MAPE: mean absolute percentage error.

## Data Availability

The datasets generated and analyzed during the current study were obtained from the Health Commission of Henan Province (http://wsjkw.henan.gov.cn/, accessed on 1 April 2021).

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
