# Peer review of "A Combined Model of SARIMA and Prophet Models in Forecasting AIDS Incidence in Henan Province, China"

_ijerph, 2022, doi:10.3390/ijerph19105910_

Round 1
Reviewer 1 Report
The authors combine a seasonal autoregressive integrated moving average (SARIMA) model with the Prophet model (a Facebook prediction tool for time series) in order to predict the incidence of AIDS in Henan province (China). The combined model is given by the weighted geometric mean of the predictions given by the two models, where the weighting coefficients are non-negative and normal. The values of such coefficients are obtained by minimizing the sum of the norms, either L1 or L2, of the logarithmic prediction errors over the observation period. The prediction performances of the single models (different kinds of SARIMA models and the Prophet model) and of the two versions of the combined model (L1-based and L2-based) are analysed. The L1-based model combining SARIMA(1,0,1)(0,1,1)[12] and Prophet shows the minimal (mean square, mean absolute, mean percentage absolute) error in predicting the AIDS incidence data of Henan province.
I believe that the paper does not provide a significant contribution to the mathematical modelling of AIDS epidemic. No innovative method is introduced in the present analysis and the relevance of the study is limited to the prediction of the specific data examined in the paper. Moreover, the English writing of many sentences is quite poor and awkward.
In order to make the paper acceptable for publication in the “International Journal of Environmental Research and Public Health”, I strongly suggest to fix the following three major points:
- the innovation of the proposed model must be examined in depth, extending the discussion and conclusion sections. The authors should highlight the advantage of using their mathematical approach instead of more standard mechanistic models (such as ODE, PDE, SDE);
- the description of the SARIMA models and of the Prophet model must be extended. More details related to the mathematical structure of the two types of models must be explicitly given in the related sections;
- an extensive writing editing by means of an English native speaker is required.
Minor issues are also reported in the following list.
Minor issues:
- 1, line 35: leave an empty space after “October 2020.”;
- 1, line 36: “…and specific drugs can be used…”;
- 2, lines 55, 56: the sentence “…combining different models can increase the chance of capturing various” seems incomplete;
- 4, line 117 (Eq. (3)): the product becomes a sum after the log transformation;
- 4, line 119: maybe you mean “the weighted geometric average of the predicted values xit”;
- 4, lines 124 and 128: the definition of the prediction error is a bit confusing; more precisely, you should say “the logarithmic error between the actual value at time t and the corresponding predicted value of the ith single model”;
- 4, line 129 (Eq. (6)): the coefficients “li” are missing in the sum of the errors “eit”;
- 4, line 136; Pag. 8 line 204: you should mention “model (1)-(2)” instead of “model (1)”;
- Figure 4: the figure seems to refer to the model prediction from the beginning of 2012 to the end of 2020. However the caption says “…from January 2020 to December 2020”. Moreover, the sentence from line 186 to 188 of pag. 7 seems to confirm that the prediction period is the same one declared in the caption (“…from January to December 2020”). Please fix the inconsistency.
I also suggest to introduce a legend that clearly state which quantities are represented by the graphical objects (solid lines, black dots, shadow area etc);
- 8, line 207: “…was an optimal combinatorial model, as it is F(l1, l2)<min{F1, F2}”;
- 9, line 219: “…and two combined models…”.
Author Response
Response to Reviewer 1 Comments
Point 1: The innovation of the proposed model must be examined in depth, extending the discussion and conclusion sections. The authors should highlight the advantage of using their mathematical approach instead of more standard mechanistic models (such as ODE, PDE, SDE)
Response 1: Your suggestion is constructive. According to your comment, we have revised the discussion of this study, and further elaborated the innovation of the proposed model in this study. Based on previous studies, we combined SARIMA model and Prophet model based on L1 norm to predict the monthly incidence of AIDS in Henan Province. We found that the prediction effect of Prophet model was better than that of SARIMA model, the combined model was better than single model, and compared with the L2 norm, the L1 norm could improve the prediction accuracy of the combined model. In addition, we explored the reasons for this result. This indicated that the combined model of SARIMA and Prophet model based on L1 norm is a suitable method to predict the incidence of AIDS in Henan Province, and our findings could provide theoretical evidence for the government to formulate policies regarding AIDS prevention. (line 252-331, page 10-11). At the same time, we have also modified the corresponding parts of the introduction and abstract.
Point 2: The description of the SARIMA models and of the Prophet model must be extended. More details related to the mathematical structure of the two types of models must be explicitly given in the related sections.
Response 2: According to your comment, we have added two paragraphs in methods. On the one hand, we added the modeling steps of time series in SARIMA model section. (line 108-117, page 3) On the one hand, we added the mathematical structure of the trend model in Prophet model section.(line 131 to 137, page 4)
Point 3: An extensive writing editing by means of an English native speaker is required.
Response 3: According to your comment, we have had the manuscript polished with an English speaker assistance in writing.
Point 4: 1, line 35: leave an empty space after “October 2020.”
Response 4: According to your comment, we have corrected the sentence. (line 45, page 1)
Point 5: 1, line 36: “…and specific drugs can be used…”;
Response 5: According to your comment, we have corrected the sentence. (line 46, page 2)
Point 6: 2, lines 55, 56: the sentence “…combining different models can increase the chance of capturing various” seems incomplete;
Response 6: According to your comment, we have revised the sentence. (line 66-67, page 2)
Point 7: 4, line 117 (Eq. (3)): the product becomes a sum after the log transformation;
Response 7: According to your comment, we have corrected it.
Point 8: 4, line 119: maybe you mean “the weighted geometric average of the predicted values ”;
Response 8: According to your comment, we have corrected the sentence. (line 153, page 4)
Point 9: 4, lines 124 and 128: the definition of the prediction error is a bit confusing; more precisely, you should say “the logarithmic error between the actual value at time t and the corresponding predicted value of the ith single model”;
Response 9: According to your comment, we have corrected the sentence. (line 158, page 4)
Point 10: 4, line 129 (Eq. (6)): the coefficients “” are missing in the sum of the errors “”;
Response 10: According to your comment, we have added to equation (6).
Point 11: 4, line 136; Pag. 8 line 204: you should mention “model (1)-(2)” instead of “model (1)”;
Response 11: Since model (1) is a whole, sꞏt is the limiting condition for the value of in , so we retained the expression of “model (1)”. (line 240, page 9)
Point 12: Figure 4: the figure seems to refer to the model prediction from the beginning of 2012 to the end of 2020. However, the caption says “…from January 2020 to December 2020”. Moreover, the sentence from line 186 to 188 of page. 7 seems to confirm that the prediction period is the same one declared in the caption (“…from January to December 2020”). Please fix the inconsistency.
I also suggest to introduce a legend that clearly state which quantities are represented by the graphical objects (solid lines, black dots, shadow area etc.);
Response 12: On the one hand, we have changed the title of figure 4 and added the legend(line 225, page 8). On the other hand, we have modified the description of Prophet model in the result section(line 219-223, page 8).
Point 13: 8, line 207: “…was an optimal combinatorial model, as it is F(l1, l2)<min{F1, F2}”;
Response 13: According to your comment, we have corrected the sentence. (line 243, page 9)
Point 14: 9, line 219: “…and two combined models…”.
Response 14: According to your comment, we have corrected the sentence.
(line 255, page 9)

Reviewer 2 Report
The authors proposed a combined model of SARIMA and Prophet model based on L1 norm to predict the incidence of AIDS in Henan. This work is novel to the best of my knowledge and I think after answering the queries below, it is fit to appear in this journal.
- I do not think authors should cite in the abstract. Also, in the abstract, it is better for authors should report a model that performed better base on the evaluation metrics rather than stating the values of MSE, MAE or MAPE. I suggest the abstract should be rewritten.
- The grammatical structure in lines 55 and 56 is wrong. Please re-read the entire article and correct all errors.
- Any comment on residual?
- Which of the parameter(s) in the model is sensitive?
- How did you avoid overfitting or false positive?
- What are your comments on the effect of seasonality?
- I also want to suggest more recent work in the direction of the article subject as follows:
https://doi.org/10.28919/cmbn/5144
https://doi.org/10.3390/healthcare9101247
DOI: 10.48129/kjs.splcov.14501
https://doi.org/10.1016/j.rinp.2021.104098
https://doi.org/10.1016/j.rinp.2021.104463
https://doi.org/10.3389/fams.2021.786983
Future research direction may be shown in Conclusion section.
Author Response
Response to Reviewer 2 Comments
Point 1: I do not think authors should cite in the abstract. Also, in the abstract, it is better for authors should report a model that performed better base on the evaluation metrics rather than stating the values of MSE, MAE or MAPE. I suggest the abstract should be rewritten.
Response 1: According to your comment, we have revised the abstract of this study. We have enriched the results section and deleted the description of stating the values. (line 11-28, page 1)
Point 2: The grammatical structure in lines 55 and 56 is wrong. Please re-read the entire article and correct all errors.
Response 2: According to your comment, we have corrected the incorrect grammar and checked the full text.
Point 3: Any comment on residual?
Response 3: For SARIMA model, checking the residual sequence of the model is one of the steps to establish the model. In this study, Ljung-box Q Test was performed on the residual sequence of candidate models. The results showed that the SARIMA (1,0,1)(0,1,1)[12] model was the best one, and passed the Ljung–Box Q Test (P=0.420). That is, there is no autocorrelation of the residual sequence, and the model is appropriate. These contents have been presented in the results(line 204-207, page 6). For prophet model, εt is the error term and is assumed to be normally distributed in this study.
Point 4: Which of the parameter(s) in the model is sensitive?
Response 4: The parameters involved in SARIMA model include d, p, P and q, Q. The selection of parameters is one of the steps to build the SARIMA model, which we added to the methods section(line 108-117, page 3). Firstly, we used the Augmented Dickey–Fuller (ADF) unit-root test to estimate whether the time series is stationary or not. Secondly, the temporal correlation structure of the transformed data is identified by examining its autocorrelation (ACF) and partial autocorrelation (PACF) functions. In addition, the values of p, P and q, Q are finally determined considering that smaller Akaike information criterion (AIC) and Bayesian information criterion (BIC) values correspond to a higher prediction accuracy. Finally, in order to test the normality of the SARIMA residuals, a white noise test is conducted in the residual series. In addition, Prophet model was used to automatically fit the data and the default parameters were mentioned in methods section(line 141-143, page 4).
Point 5: How did you avoid overfitting or false positive?
Response 5: In this study, we avoided overfitting by dividing training set and test set. The monthly incidence of AIDS in Henan province from January 2012 to December 2019 were set as the training set, and the incidence of AIDS from January to December 2020 were set as the test set. SARIMA model, Prophet model and two combined models were adopted to fit the monthly incidence of AIDS by using the data of the training set. The forecasting performances of the four fitted models were verified by using the data of the test set. The results of the training set and the test set are consistent, that is, there is no overfitting in this study. In addition, we adopted two regularization methods (L1 norm and L2 norm) to prevent overfitting.
Point 6: What are your comments on the effect of seasonality?
Response 6: About my comments on seasonality was mentioned in the second paragraph of the discussion (line 270-277, page 10). As for the incidence of AIDS reported showed seasonality, we used the SARIMA and Prophet models to eliminate seasonal trends and made the prediction more accurate.
Point 7: I also want to suggest more recent work in the direction of the article subject as follows: https://doi.org/10.28919/cmbn/5144; https://doi.org/10.3390/healthcare9101247; DOI:10.48129/kjs.splcov.14501; https://doi.org/10.1016/j.rinp.2021.104098; https://doi.org/10.1016/j.rinp.2021.104463; https://doi.org/10.3389/fams.2021.786983.
Future research direction may be shown in Conclusion section.
Response 7: Thank you for these articles in the comments. These articles mentioned some new mathematical models to investigate diseases such as coronavirus disease (COVID-19), and typhoid fever. After careful reading, we quoted some articles and added some sentences in the discussion.
In the fourth paragraph of the discussion, we quoted the article of Adesoye Idowu Abioye et.al and then described our study (line 296-298, page 10). In the last paragraph of the discussion, we quoted the articles of Oshinubi K, et.al and described the future research direction (line 328-331, page 10).
The references are as follows:
- Adesoye Idowu Abioye M D U, Peter O J, Edogbanya H O, Oguntolu F A, Kayode O, Amadiegwu S. Forecasting of COVID-19 pandemic in Nigeria using real statistical data. Commun. Math. Biol. Neurosci. 2021. 2021; doi:10.28919/cmbn/5144.
- Oshinubi K, Rachdi M, Demongeot J. Modeling of COVID-19 Pandemic vis-à-vis Some Socio-Economic Factors. 2022;7. 2022; doi:10.3389/fams.2021.786983.
- Oshinubi K, Rachdi M, Demongeot J. Analysis of Reproduction Number R0 of COVID-19 Using Current Health Expenditure as Gross Domestic Product Percentage (CHE/GDP) across Countries. 2021;9:1247. 2021; doi:10.3390/healthcare9101247.

Reviewer 3 Report
Dear Authors
Thank you so much for your submission and contribution. I believe my suggestion will enhance the quality of your manuscript. My major comments are below:
- In introduction, please explain with recent literature on your applied model in this study.
- You have used Bayesian modeling to infer various seasonal patterns combined with unpredictable changepoints and wrap them in a Generalized Additional Model. So please describe these points in introduction with some sentences.
- In introduction, can you add a paragraph for the key player that leads to increase HIV infection in Henan.
- Please describe more on non-linear trends of your data that are fit with yearly, weekly, and daily seasonality, plus holiday effects more clearly with respective figure or graph.
- For the combined model based on L2 norm, we obtained ?1=0.548 and ?2=0.452. For the combined model based on L1 norm, we obtained ?1=0.4587 and ?2=0.5417, ………….Please explain why L1 and L2 norm values are just opposite each other???
Author Response
Response to Reviewer 3 Comments
Point 1: In introduction, please explain with recent literature on your applied model in this study.
Response 1: According to your comment, we have explained with recent literature on our applied model in this study and have added some sentences in introduction (line 77-81, page 2).
Point 2: You have used Bayesian modeling to infer various seasonal patterns combined with unpredictable changepoints and wrap them in a Generalized Additional Model. So please describe these points in introduction with some sentences.
Response 2:According to your comment, we have added some sentences in introduction(line 58-60, page 2).
Point 3: In introduction, can you add a paragraph for the key player that leads to increase HIV infection in Henan.
Response 3: According to your comment, we have added the key player that leads to increase HIV infection in Henan Province in introduction(line 37-43, page 2).
Point 4: Please describe more on non-linear trends of your data that are fit with yearly, weekly, and daily seasonality, plus holiday effects more clearly with respective figure or graph.
Response 4: Since the data we analyzed were the monthly incidence data of AIDS, the decomposition components of Prophet model only included trend effect and yearly, effect, which could not reflect the weekly effect and short-term holiday effect. This is also a limitation of this study. We have added these sentences in the discussion(line 322-325, page 11).
Point 5: For the combined model based on L2 norm, we obtained ?1=0.548 and ?2=0.452. For the combined model based on L1 norm, we obtained ?1=0.4587 and ?2=0.5417. Please explain why L1 and L2 norm values are just opposite each other?
Response 5:It is universally agreed that combining different models can increase the chance of capturing various and improve the prediction accuracy. The combined model based on L2 norm is based on the minimum sum of the squares of the prediction error. However, the prediction error would be enlarged or reduced after the prediction error is squared, that is, if the absolute value of prediction absolute error is greater than 1, it is larger after it is squared. If the absolute value of the predicted relative error is less than 1, it is smaller after it is squared. The combined model based on L1 norm uses the sum of the absolute values of the prediction errors. Based on the L1 norm, model is sparse and regularized. The L1 norm decreases very quickly for small weights and slowly for large weights. Perhaps as for L1 norm is better than L2 norm, the weights of SARIMA model and Prophet model are different in the two combined models. We have added these sentences to the discussion(line 310-317, page 10).
In addition, according to the results of our study, the prediction effect of Prophet model was better than that of SARIMA model. Therefore, the weight of Prophet model in the combined model should be higher, which also showed that the L1 norm adopted in this study was better.
The reference is as follows:
1. Wang B, Jiang Q, Jiang P. A combined forecasting structure based on the L(1) norm: Application to the air quality. Journal of environmental management. 2019;246:299-313. 2019; doi:10.1016/j.jenvman.2019.05.124.

Round 2
Reviewer 1 Report
The major issues have been basically addressed in the revised manuscript.
The paper can be accepted for publication.
Minor point:
- Pag. 2, line 80: the sentence "...combining different models can increase the chance of capturing various". Various what?!